# Mass Spectrometry Data Repository Enhances Novel Metabolite Discoveries with Advances in Computational Metabolomics

**DOI:** 10.3390/metabo9060119

**Published:** 2019-06-24

**Authors:** Hiroshi Tsugawa, Aya Satoh, Haruki Uchino, Tomas Cajka, Makoto Arita, Masanori Arita

**Affiliations:** 1Metabolome informatics research team, RIKEN Center for Sustainable Resource Science, Yokohama 230-0045, Japan; aya.hayaishi@riken.jp; 2Laboratory for metabolomics, RIKEN Center for Integrative Medical Sciences, Yokohama 230-0045, Japan; haruki-uchino@keio.jp (H.U.); makoto.arita@riken.jp (M.A.); 3Division of Physiological Chemistry and Metabolism, Graduate School of Pharmaceutical Sciences, Keio University, Minato-ku, Tokyo 105-8512, Japan; 4Department of Metabolomics, Institute of Physiology of the Czech Academy of Sciences, Videnska 1083, 14220 Prague, Czech Republic; tomas.cajka@fgu.cas.cz; 5Department of Translational Metabolism, Institute of Physiology of the Czech Academy of Sciences, Videnska 1083, 14220 Prague, Czech Republic; 6Cellular and Molecular Epigenetics Laboratory, Graduate School of Medical Life Science, Yokohama City University, Tsurumi, Yokohama 230-0045, Japan; 7National Institute of Genetics, Mishima 411-8540, Japan

**Keywords:** data repository, computational metabolomics, reanalysis, lipidomics, data processing

## Abstract

Mass spectrometry raw data repositories, including Metabolomics Workbench and MetaboLights, have contributed to increased transparency in metabolomics studies and the discovery of novel insights in biology by reanalysis with updated computational metabolomics tools. Herein, we reanalyzed the previously published lipidomics data from nine algal species, resulting in the annotation of 1437 lipids achieving a 40% increase in annotation compared to the previous results. Specifically, diacylglyceryl-carboxyhydroxy-methylcholine (DGCC) in *Pavlova lutheri* and *Pleurochrysis carterae*, glucuronosyldiacylglycerol (GlcADG) in *Euglena gracilis,* and *P. carterae*, phosphatidylmethanol (PMeOH) in *E. gracilis*, and several oxidized phospholipids (oxidized phosphatidylcholine, OxPC; phosphatidylethanolamine, OxPE; phosphatidylglycerol, OxPG; phosphatidylinositol, OxPI) in *Chlorella variabilis* were newly characterized with the enriched lipid spectral databases. Moreover, we integrated the data from untargeted and targeted analyses from data independent tandem mass spectrometry (DIA-MS/MS) acquisition, specifically the sequential window acquisition of all theoretical fragment-ion MS/MS (SWATH-MS/MS) spectra, to increase the lipidomic annotation coverage. After the creation of a global library of precursor and diagnostic ions of lipids by the MS-DIAL untargeted analysis, the co-eluted DIA-MS/MS spectra were resolved in MRMPROBS targeted analysis by tracing the specific product ions involved in acyl chain compositions. Our results indicated that the metabolite quantifications based on DIA-MS/MS chromatograms were somewhat inferior to the MS^1^-centric quantifications, while the annotation coverage outperformed those of the untargeted analysis of the data dependent and DIA-MS/MS data. Consequently, integrated analyses of untargeted and targeted approaches are necessary to extract the maximum amount of metabolome information, and our results showcase the value of data repositories for the discovery of novel insights in lipid biology.

## 1. Introduction

Many studies using mass spectrometry (MS)-based untargeted metabolomics have provided novel insights in biology, and the importance of metabolomics data repositories has been recognized [1]. In addition to the international data repositories, including Metabolomics Workbench [2], MetaboLights [3], and GNPS MassIVE [4], institute-oriented repositories such as RIKEN DropMet (http://prime.psc.riken.jp/) are available for sharing raw MS data. MS repositories aim to (A) increase the transparency and reproducibility of MS-centric metabolomics studies, (B) provide a benchmark for testing new analytical and computational methodologies, (C) share the results of metabolome analyses for providing opportunities for data-driven hypothesis generation, and (D) reanalyze published data with continuous identification efforts to obtain novel insights and discover novel metabolites. Nevertheless, few studies have demonstrated the value of MS repositories except for the first purpose. Case studies showing remarkable results toward the other three purposes would facilitate data sharing by researchers and academic journals [5]. 

We published nine algal lipidomics datasets in 2015 which are available on the RIKEN DropMet. According to previous reports, 1023 lipids were annotated by integrating the results of data-dependent MS/MS acquisition (DDA-MS/MS) and data-independent MS/MS acquisition (DIA-MS/MS) with an in-silico MS/MS spectral library of lipids to assign the algal phylogenetic tree based on lipid properties [6]. Although comprehensive lipid analysis has been achieved by these previous methodologies, the coverage of algal lipids can be improved by two major methodological updates. First, the count of annotated lipids can be increased by updating the in-silico spectral library with continuous data curation efforts, where 1,051,894 spectra of 525,947 molecules from 90 lipid classes are currently registered in MS-DIAL (version 3.68), while 122,844 spectra of 61,422 molecules from 24 lipid classes were registered in the first version of MS-DIAL (version 1.82) [6,7,8]. Herein, the characterization of 10 newly incorporated lipid classes including lysophosphatidylserine (LPS), lysophosphatidylglycerol (LPG), phosphatidylmethanol (PMeOH), glucuronosyldiacylglycerol (GlcADG), diacylglyceryl-carboxyhydroxy-methylcholine (DGCC), and its lyso-type form (LDGCC), as well as oxidized fatty acids containing phosphatidylcholine (OxPC), phosphatidylethanolamine (OxPE) phosphatidylglycerol (OxPG), and phosphatidylinositol (OxPI) are highlighted for algal lipid profiling. Second, the peak capacity in LC-MS/MS data could be increased by the integrated analyses of MS-DIAL-based untargeted [6,9] and MRMPROBS-based targeted analyses [10,11] for DIA-MS/MS data. In DIA-MS/MS data, MS-DIAL uses the MS^1^ chromatogram trace for metabolite quantification and the deconvoluted MS/MS spectrum for metabolite annotation. Therefore, the program does not completely resolve co-eluted metabolites, as only a singlet MS^1^ chromatogram peak from mixed metabolite ions can be obtained [12]. On the other hand, MRMPROBS can use either the MS^1^ or MS/MS chromatogram for metabolite quantification and metabolite diagnostics is performed by the integrated score of the peak groups from the user-defined precursor-product transitions library [10]. Notably, co-eluted metabolites in the MS^1^ chromatogram trace can be resolved using the MS/MS chromatograms which differ based on the unique product ions from the lipid structure, resulting in increased peak capacity, i.e., increased deconvolution efficiency, in the LC-MS/MS dataset. 

Herein, we showcase novel lipid discoveries using algal lipidomics data as a benchmark. We discovered novel lipid classes which have never been reported in the algal species using updated in-silico MS/MS libraries. Moreover, we demonstrate increased annotated lipid coverage by integrating the pipelines of the MS-DIAL and MRMPROBS programs. Although the integrated pipeline can be executed using other state-of-the-art program combinations, such as XCMS [13] and MZmine 2 [14], for untargeted analysis and MetDIA for targeted analysis [15], the selected programs support the direct link from untargeted to targeted analyses with a user-friendly graphical user interface (GUI) where manual peak-picking required for targeted approaches is acceptable. This study highlights the importance of mass spectrometry data repositories to deepen our understanding of lipids in algal species. 

## 2. Materials and Methods

### 2.1. Overview of Data Analysis Workflow

Since the MetaboLights database and repository was launched in 2012 by the European Bioinformatics Institute (EMBL-EBI) as the first repository for metabolomics data, data submission has continuously increased (~2.5 TB data was available in June 2019), and accessibility and awareness have been enhanced through the efforts of MetabolomeXchange (http://www.metabolomexchange.org) and OmicsDI [16]. RIKEN DropMet (http://prime.psc.riken.jp/menta.cgi/prime/drop_index) has also been launched in 2009 to share MS-based metabolomics data from RIKEN, in which ~300 GB of data from 29 studies are currently available; a part of this repository, i.e., the algae lipidomics data, was used in this study. 

On the other hand, data processing tools like MS-DIAL [6,9], XCMS [13], and MZmine 2 [14] have continuously been updated with database curations like Metlin [17] in XCMS. Since the LipidBlast library was released in 2013 as the first public in-silico library for lipids [18], the fork libraries for quadrupole/time-of-flight mass spectrometry (QTOF-MS) with collision-induced dissociation (CID) and orbital ion trap MS with higher-energy collisional dissociation (HCD) data have been developed in MS-DIAL [6,7] owing to the continuous effort. The annotation described in this study can be executed in the MS-DIAL version 3.66 or higher. All programs, i.e., MS-DIAL, MRMPROBS, and the related spectral libraries are available on the RIKEN PRIMe website (http://prime.psc.riken.jp/).

### 2.2. Mass Spectrometry Data

The DDA and DIA lipidomics data obtained in positive and negative ion modes of nine algal species were downloaded from the RIKEN DropMet website (http://prime.psc.riken.jp/menta.cgi/prime/drop_index; ID, DM0022). Briefly, the extraction of algal lipids was performed using a biphasic solvent system of cold methanol, methyl *tert*-butyl ether (MTBE), and water followed by lipid separation via reversed-phase liquid chromatography. Both DDA and DIA data were acquired using a QTOF mass spectrometer (TripleTOF 5600+, SCIEX). For DIA (SWATH-MS/MS), a 21 Da isolation window was used for selecting precursor ions shifting over an *m*/*z* 100–1250 mass range. Further details are provided in the previous study [6]. 

### 2.3. Software Programs

MS-DIAL version 3.06 and MRMPROBS version 2.44 were used herein. All programs including the latest version are freely available on the RIKEN PRIMe website (http://prime.psc.riken.jp/). 

The same parameters in MS-DIAL were used for DDA and DIA data analyses: retention time begin, 0 min; retention time end, 100 min; mass range begin, 0 Da; mass range end, 5000 Da; accurate mass tolerance (MS1), 0.01 Da; MS2 tolerance, 0.025 Da; maximum charge number, 2; smoothing method, linear weighted moving average; smoothing level, 3; minimum peak width, 5 scan; minimum peak height, 1000; mass slice width, 0.1 Da; sigma window value, 0.5; MS2Dec amplitude cut-off, 0; exclude after precursor, true; keep isotope until, 0.5 Da; keep original precursor isotopes, false; exclude after precursor, true; retention time tolerance for identification, 4 min; MS1 for identification, 0.01 Da; accurate mass tolerance (MS2) for identification, 0.05 Da; identification score cut-off, 70%; using retention time for scoring, true; relative abundance cut off, 0; top candidate report, true; retention time tolerance for alignment, 0.05 min; MS1 tolerance for alignment, 0.015 Da; peak count filter, 0; remove feature based on peak height fold-change, true; sample max/blank average, 5; keep identified and annotated metabolites, true; keep removable features and assign the tag for checking, true; replace true zero values with 1/10 of the minimum peak height over all samples, false. Lipid annotation was performed automatically using the in-silico MS/MS spectral library described below, and the result was manually curated with the confirmation of the characteristic product ions and neutral losses to reduce false-positive annotations. 

The parameters in MRMPROBS were set as follows: MS1 tolerance, 0.01 Da; MS2 tolerance, 0.025 Da; smoothing method, linear weighted moving average; smoothing level, 1; minimum peak width, 5 scan; minimum peak height, 200; retention time tolerance for identification, 0.1 min; amplitude tolerance for identification, 15%; minimum posterior, 70%; the abundance ratios in reference library were automatically generated by MS-DIAL. The results of metabolite annotation and peak picking were manually curated using the graphical user interface of MRMPROBS. 

### 2.4. In-Silico MS/MS Spectral Libraries

The diagnostic ions used to characterize lipid classes were determined using authentic standards, experimental MS/MS spectra of biological samples, or MS/MS spectral information reported in the literature. The MS/MS spectra of PMeOH, LPS, and LPG were confirmed using the standard compounds PMeOH 16:0–16:0, LPG 18:1, and LPS 18:1 (Avanti Polar Lipids, Inc., Alabaster, AL, USA). The DGCC and LDGCC spectra were examined in the DDA-MS/MS data of *Pavlova lutheri* because these lipids were previously discovered in *P. lutheri* [19] and the corresponding literature’s MS/MS spectrum was utilized to create an in-silico MS/MS library [20]; the MS/MS spectra that have electronically been described in a peer-review journal but not recorded in publicly and commercially available spectral databases such as MassBank and NIST are referred to as the literature’s MS/MS. The in-silico MS/MS spectral libraries for oxidized phospholipids were developed considering our previously published data [21]. The library creation for GlcADG was based on the literature’s MS/MS spectrum [22]. Information regarding ion abundances in the MS/MS spectral libraries was based on our LC-MS/MS experimental conditions and the detailed analytical conditions were described in a previous study [7]. Briefly, the MS data were acquired in information-dependent mode (IDA), i.e., DDA, using SCIEX TripleTOF 5600+ or 6600 systems. The mass range, collision energy, and collision energy spread were set to *m*/*z* 70–1250, 45 V, and 15 V, respectively. 

## 3. Results and Discussion

### 3.1. Novel Lipid Characterizations in Algae with Enriched In-Silico Spectral Libraries

The global lipid profiling of nine algal lipids was achieved in 2015 and 15 lipid classes were characterized [6]. These classes include free fatty acid (FFA), di- and triacylglycerols (DAG and TAG), seven phospholipid classes (phosphatidylcholine, PC; phosphatidylethanolamine, PE; phosphatidylglycerol, PG; phosphatidylinositol, PI; phosphatidylserine, PS; lysophosphatidylcholine, LPC; and lysophosphatidylethanolamine, LPE), mono- and digalactosyldiacylglycerol (MGDG and DGDG), sulfoquinovosyldiacylglycerol (SQDG), diacylglyceryltrimethylhomoserine (DGTS), and its lyso-type form (LDGTS). Of these, the most common lipid classes in the photosynthetic membranes of plants, cyanobacteria, and algae, which include PG, SQDG, MGDG, and DGDG, have been characterized in all algal species. In contrast, *N*,*N*,*N*-trimethylammonium cation-containing lipids, i.e., PC and DGTS, were characterized as species-specific lipid classes. For example, *Chlamydomonas reinhardtii* only contains DGTS, while Chlorella species only contain PC as their characteristic positively charged membrane lipids. Since the specificity of lipid metabolism is highly influenced by genetics, evolution, and the environment of living organisms, increasing lipidomics coverage is an emerging requirement in biology.

Herein, 17 lipid classes were newly characterized in algal species using enriched MS/MS spectral libraries and include LPS, LPG, oxidized phospholipids (OxPC, OxPE, OxPG, OxPI), seven ceramide classes (Cer-NS, Cer-NDS, Cer-AP, Cer-NP, Cer-AS, Cer-ADS, and HexCer-AP), PMeOH, GlcADG, DGCC, and LDGCC (Table 1). The diagnostic ions are summarized in Table 2, and the annotation strategy and nomenclature of the ceramides are reported elsewhere [7]. The phytoceramide species, Cer-AP and Cer-NP, were characterized in all algal species, while several lipid classes were determined to be algal species-specific (Figure 1).

For example, DGCC and LDGCC lipids were only characterized in *Pavlova lutheri* and *Pleurochrysis carterae*. DGCC is well-known as a major betaine lipid of non-plastid membranes in *P. lutheri* [19], while this lipid class has never been reported in *P. carterae*. Thus, further investigation in *P. carterae* is required to define its exact stereochemistry. The MS-based lipidomics platform does not resolve the stereochemistry of acyl chains and sometimes lipid classes cannot uniquely be assigned, although a large variety of lipid molecules can be covered by tracing the lipid class-specific product ions and neutral losses. For example, DGTS and diacylglyceryl hydroxymethyl-*N*,*N*,*N*-trimethyl-β-alanine (DGTA), the major lipid class in *P. lutheri*, are characterized by the same diagnostic ions (*m*/*z* 144.102 and *m*/*z* 236.149) [23] under our experimental conditions, so the annotation must be determined by considering the genetic background in mass spectrometry-based metabolite annotations [9]. GlcADG, also known as diacylglyceryl glucuronide (DGGA), was observed in *Euglena gracilis*, *P. lutheri*, and *P. carterae*. GlcADG is known to be accumulated in response to phosphorus starvation in *Arabidopsis thaliana* and *Oryza sativa* [24], and this lipid class is commonly observed in several algal species. A previous study has also reported its existence in *P. lutheri*. Furthermore, the rare phospholipid PMeOH class was characterized in *E. gracilis*, although it could also be detected as an artifact of extraction [25]. Finally, several oxidized fatty acid-containing phospholipids, including OxPC, OxPE, OxPG, and OxPI, were characterized in *Chlorella variabilis*. Although further investigation of these discovered lipids is required to determine whether the lipid class is endogenously biosynthesized in a specific algal species [26], our results indicated that the reanalysis of the published data with the updated annotation workflow could provide new insights and hypotheses not previously reported.

### 3.2. Strategy to Link Untargeted- and Targeted Analyses for Increasing Lipid Coverage

We further demonstrated the increased lipid profiling coverage by integrating untargeted and targeted analysis approaches (Figure 2). Although MS-DIAL involves a deconvolution algorithm, MS2Dec, to process the DIA-MS/MS data, MS2Dec requires at least two data point peak-top differences of co-eluted peaks for chromatogram deconvolution [6,12]. Moreover, MS-DIAL uses MS^1^ chromatogram traces for metabolite quantification, while the MS/MS chromatogram can effectively be used to annotate and quantify the target metabolite in DIA-MS/MS data [15]. Therefore, the MRMPROBS program was used, in which the user-friendly GUI was available for data curation in addition to its favorable algorithm aspects [10,11], to compensate for the shortcomings of the MS-DIAL program. First, both DDA- and DIA-MS/MS data were analyzed using MS-DIAL. Second, all spectrometrically ‘matched’ candidates to an MS/MS spectrum were obtained and the function was newly developed. In this study, we utilized the DDA-MS/MS spectral data to examine the co-eluted metabolite profile because of the better spectrum quality than that of the DIA-MS/MS spectra. Third, the reference format file, which contains (1) metabolite name (2) retention time, (3) precursor ion *m*/*z*, and product ion *m*/*z* list, and (4) ion abundance ratios, was generated to cover all matched candidates. Finally, the DIA-MS/MS data were analyzed by MRMPROBS using the reference library where the peak left and right edges of the metabolite peak in each MS/MS chromatogram trace were manually refined. 

Importantly, smoothing level 1 was used in the MRMPROBS program and it was set to 3 in the MS-DIAL program. The higher smoothing level allows for the determination of peak left and right edges in the automated data analysis pipeline owing to the reduced noise level. Therefore, the higher smoothing value was used in the untargeted analysis. Conversely, the co-eluted peaks are often merged as a single peak. Because all chromatogram peaks could manually be checked and modified in the GUI, the lower smoothing value was used in the targeted analysis software.

### 3.3. Showcase of Newly Resolved Lipid Profiles by MS/MS-Centric Data Analysis

We showcased the methodology by profiling three co-eluted lipid molecules, DGDG 16:2–18:1, DGDG 16:1–18:2, and DGDG 16:0–18:3, with the same precursor *m*/*z* and similar retention times (Figure 3a). In the analysis of *E. gracilis*, only DGDG 16:2–18:1 was annotated correctly in MS-DIAL based on the spectral match score, while the MS/MS spectrum was partially interpreted as DGDG 16:1–18:2 and 16:0–18:3. Using the principles of MS-DIAL, the spectra of co-eluted metabolites are annotated using the representative metabolite with the highest spectral matching score, although it could be manually modified. Therefore, MS-DIAL generated the MRMPROBS reference format for the three lipid candidates, and the lipids were quantified using the MRMPORBS program. As a result, DGDG 16:0–18:3 was determined to be the major component of the co-eluted lipids in *Chlamydomonas reinhardtii*, *Auxenochlorella protothecoides*, *C. sorokiniana*, *C. variabilis*, and *Dunaliella salina*, DGDG 16:2–18:1 the major component in *E. gracilis*, and DGDG 16:1–18:2 in *Nannochloropsis oculate* and *P. carterae* (Figure 3b), where the lipid differences clearly reflected the differences in the phylum. Importantly, these differences could not be resolved using MS-DIAL untargeted analysis (Figure 3c) because the program uses MS^1^-centric peak quantification, i.e., the red traces in the chromatograms shown in Figure 3b. This indicated that the DIA-MS/MS data enabled the increased coverage of metabolic profiling, as described elsewhere [27], and the two program suites MS-DIAL and MRMPROBS provided a solution to fully utilize the information-rich MS/MS spectral data for comprehensive metabolome analyses.

### 3.4. Comparison of Untargeted- and Targeted Analysis Results

We characterized 1437 molecules in nine algal species, and the total count of annotated lipids was 40% higher than that (1023) of the lipids annotated using the previously developed methodology (Figure 4a, Table 1, Appendix A). Moreover, we examined correlations among the quantification methods, including MS^1^-centric peak height in DDA and DIA-MS/MS data, and MS/MS-centric peak height and area in the DIA-MS/MS data. As expected, the correlation between peak height and area in DIA-MS/MS data were high (Figure 4b, right-bottom), but the correlation between peak heights in DDA and DIA-MS/MS data could be affected by the mass spectrometry settings [6] where the total scan cycle times were different in DDA (650 ms) and DIA-MS/MS (730 ms) data acquisition (Figure 4b, top-left). These differences were also caused by different LC-MS analysis days, where the MS sensitivities differed for each lipid class. Surprisingly, our results indicated that the correlations between MS^1^ and MS/MS chromatograms were highly dependent on the lipid classes. This indicates that the sensitivity of the product ions is different for each lipid class, and the correlation value is high when abundances are compared within each lipid class. The dynamic range using MS/MS chromatograms was narrower than that of the MS^1^ chromatograms, and the saturation behavior was observed in the correlation plots, especially for TAG lipids, between MS^1^- and MS/MS-centric quantifications. In fact, the SWATH-MS/MS channels of the TripleTOF 5600+ instrument have a lower linear dynamic range compared to MS^1^. These results suggest that MS^1^-centric metabolite quantification is slightly superior to that of the MS/MS-centric quantification, while the annotation coverage in MS/MS-centric analyses outperformed the untargeted analysis pipeline. 

Finally, a detailed investigation of fatty acid properties revealed the uniqueness of the acyl chains in each algal species (Figure 4c). Importantly, we used the fatty acid counts included in the lipid classes instead of the ion abundances because lipid ionization efficiency is highly dependent on the lipid class and retention time. In Plantae, 16:0, 16:1, 16:2, 16:3, 18:0, 18:1, 18:2, and 18:3 are known to be major acyl chains [28], while 16:4 and 18:4 are highly distributed in Chlorophyceae including *C. reinhardtii* and *D. salina* compared to Trebouxiophyceae including Chlorella species. Moreover, the acyl chain of 18:0 is not often observed in glycerolipids and glycerophospholipids in nine algal species. In Chromista and Protozoa, polyunsaturated fatty acids (PUFAs) are enriched in addition to the common 16 and 18 carbon length series. These results show that various PUFAs, such as 20:4, 20:5, 22:5, and 22:6, were observed in *E. gracilis* while 20:5 and 22:6 were enriched in *P. lutheri*, and 20:4 and 20:5 in the lipids of *N. oculate*. These observations were achieved using the MS/MS chromatogram traces for lipid quantification, and the approach is effective for investigating lipid profiles in living organisms to deepen our understanding of lipid metabolism and its connection with gene expression and enzyme activities.

In general, untargeted analysis searches are conducted for as many metabolites as possible to generate hypotheses in biology, though the rate of false-positive annotations becomes higher than that in targeted analysis; therefore, data analysts should devote much time and effort in curating annotation results. On the other hand, the targeted analysis focuses on a limited number of metabolites with less false-positive rate, though curation is still needed to modify the peak-picking results. Although the automated pipelines with the estimation of false discovery rate (FDR) in annotations have also been developed in metabolomics [29,30,31] for large-scale datasets like cohort studies, data analysts should recognize the pitfall in annotations that may be mentioned as false-positive metabolites in the metabolome and lipidome data sheet for statistical analyses. Therefore, the integrated analysis by untargeted and targeted techniques is important to reduce misleading results in biological studies. In fact, the metabolites of interest obtained from the integrated results must be validated using authentic standard compounds and further biological experiments to compensate for the lack of current MS instruments and informatic techniques providing limited stereochemical and isomer information.

## 4. Conclusions

Consequently, the reanalysis of published data was demonstrated where 17 lipid classes were newly characterized in nine algal species in addition to the 15 lipid classes annotated previously. In effect, the coverage of lipid classes was doubled by updating the computational mass spectrometry techniques and mass spectral libraries, and our reanalysis indicates the value of MS data repositories where the raw data could be utilized as a benchmark for new software programs and data-driven hypothesis generation. The lipidomics workflow is also executable with hydrophilic interaction chromatography (HILIC) [32] or supercritical fluid chromatography (SFC) [33], in which the molecules can be separated based on the specific chemical properties of each lipid class, enabling efficient exclusion of false-positive annotations from incorrect lipid classes. Ion mobility MS provides another diagnostic criterion, viz. collision cross-section (CCS), to increase the confidence in lipid annotation [34]. Although we only showcased the increase in lipid profiling coverage, this strategy could also be applied to more diverse metabolites with experimentally acquired spectral libraries. There are three types of spectral databases: (1) completely open-access, i.e., all records are browsable and downloadable (e.g., MassBank [35], PlaSMA [9], Fiehnlib [36], and GNPS [4]), (2) limited access, i.e., browsable but not downloadable (e.g., Metlin [17] and mzCloud (https://www.mzcloud.org/)), and (3) licensed databases such as NIST and Wiley, and the integrated databases cover the MS/MS spectra of approximately 12,000 unique metabolites [9,26]. Importantly, all these databases have increasingly been updated by the continuous effort of mass spectrometrists; therefore, success similar to that obtained herein is achievable by reanalyzing public data using the upgraded databases. 

In conclusion, the science of metabolomics and lipidomics now enters a new era owing to state-of-the-art analytical techniques and informatics platforms where metabolic profiling is semi-automatically executable [37]. Therefore, MS data repositories will become increasingly important to reach a ‘standard’ in genomics and transcriptomics data sciences. Our computational workflow could be used as a pipeline for metabolomics and lipidomics data processing and the understanding of metabolism is deepened by advances in computational metabolomics. 

## Figures and Tables

**Figure 1 metabolites-09-00119-f001:**
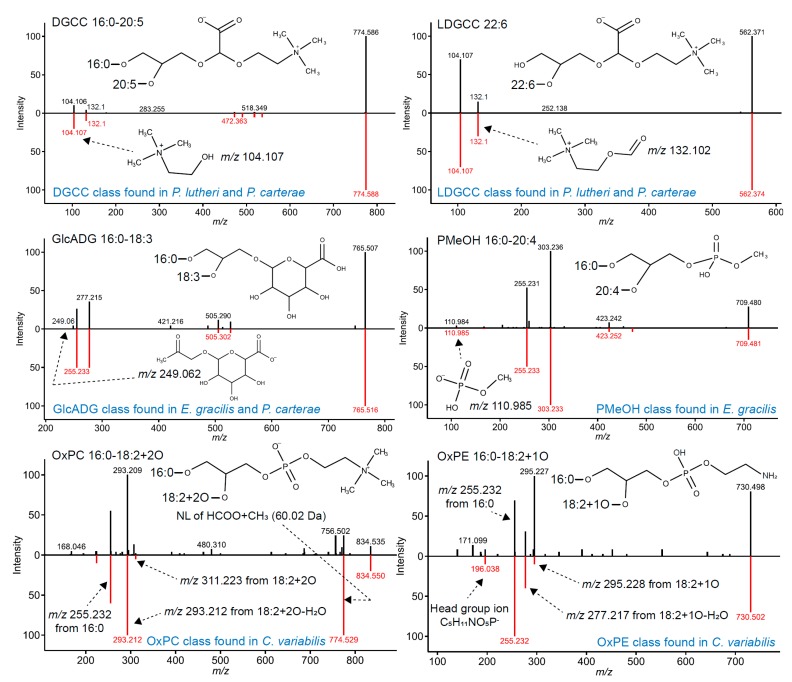
Lipid characterization using enriched in-silico MS/MS spectral libraries. The in-silico MS/MS characterizations for phosphatidylmethanol (PMeOH), glucuronosyldiacylglycerol (GlcADG), oxidized fatty acids containing phosphatidylcholine (OxPC) and phosphatidylethanolamine (OxPE), diacylglyceryl-carboxyhydroxy-methylcholine (DGCC), and its lyso-type form (LDGCC) are showcased. The upper- and bottom spectra show the experimental and in-silico MS/MS spectra, respectively. The string character indicates the abbreviation of fatty acids, and the ester link of the fatty acids is also described by string characters. NL refers to neutral loss. The algal species where the classified lipids are observed is also highlighted.

**Figure 2 metabolites-09-00119-f002:**
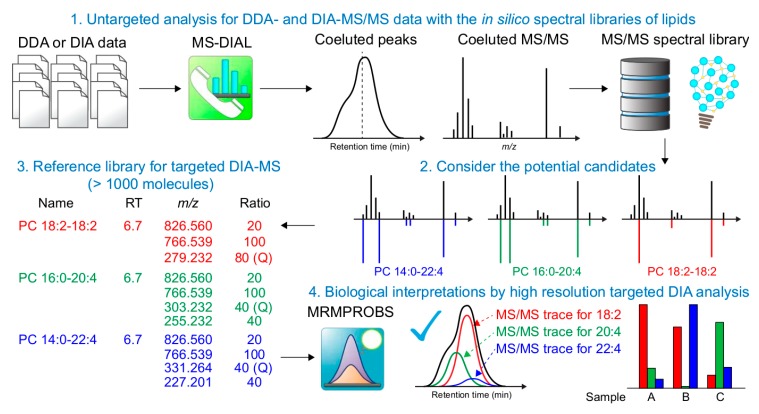
Integrated strategy of untargeted and targeted analyses to increase the coverage of annotated lipids. First, the data dependent (DDA) and independent (DIA) acquisition data are analyzed using the untargeted analysis pipeline where peak-picking, MS/MS assignment, and peak alignment are performed. Second, all potential lipid candidates that exceeded the cut-off of mass spectral similarity are obtained. Third, a reference library containing the target metabolite name, retention time, precursor ion *m*/*z*, product ion *m*/*z* list, and ion abundance ratios is automatically generated. In our study, the ratio “100” indicates that the trace is used to quantify the metabolite, and the diagnostic ion for characterizing the metabolite is described by “Q”. Finally, the MS/MS chromatograms are analyzed by the targeted analysis pipeline where the chromatogram traces are evaluated with the reference libraries combined with manual curations.

**Figure 3 metabolites-09-00119-f003:**
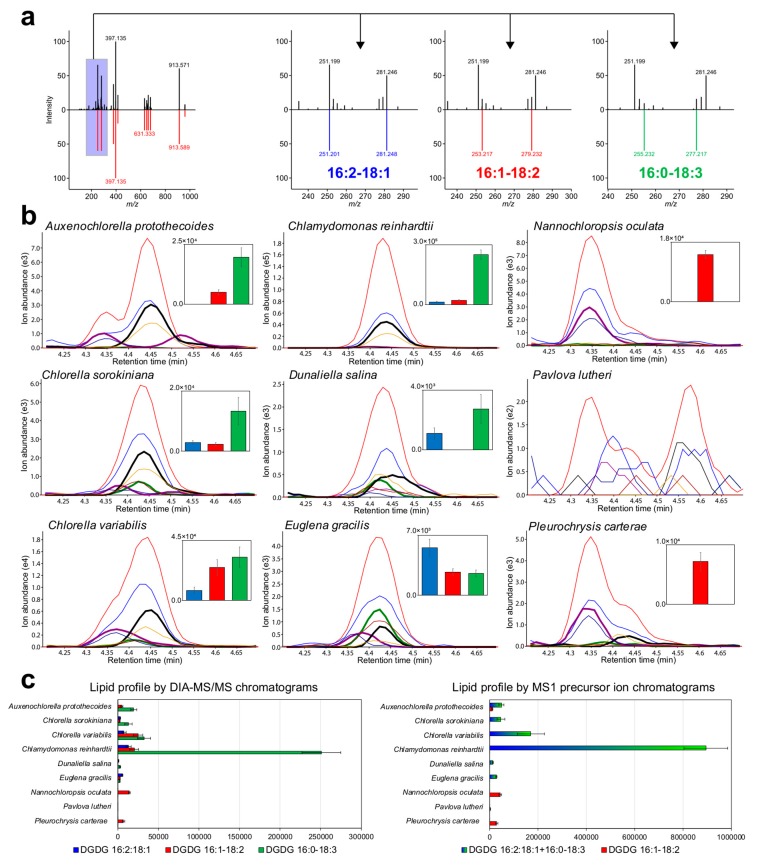
Showcase of the MS/MS-centric lipid quantifications. (**a**) The co-eluted MS/MS spectra of digalactosyldiacylglycerol (DGDG) 16:2–18:1, DGDG 16:1–18:2, and DGDG 16:0–18:3. (**b**) The red and blue traces in the chromatograms for each algal species show the extracted ion chromatograms of the precursor and product ions of *m*/*z* 397.135 used for the lipid class diagnostics, respectively. The bold green, black, and purple traces show the extracted ion MS/MS chromatograms of the *m*/*z* 251.201, *m*/*z* 253.217, and *m*/*z* 255.232 ions for fatty acids 16:2, 16:1, and 16:0, respectively. (**c**) The three molecules are quantified by the traces of unique product ion chromatograms in the targeted analysis pipeline, while two are not resolved in the MS^1^ chromatogram traces.

**Figure 4 metabolites-09-00119-f004:**
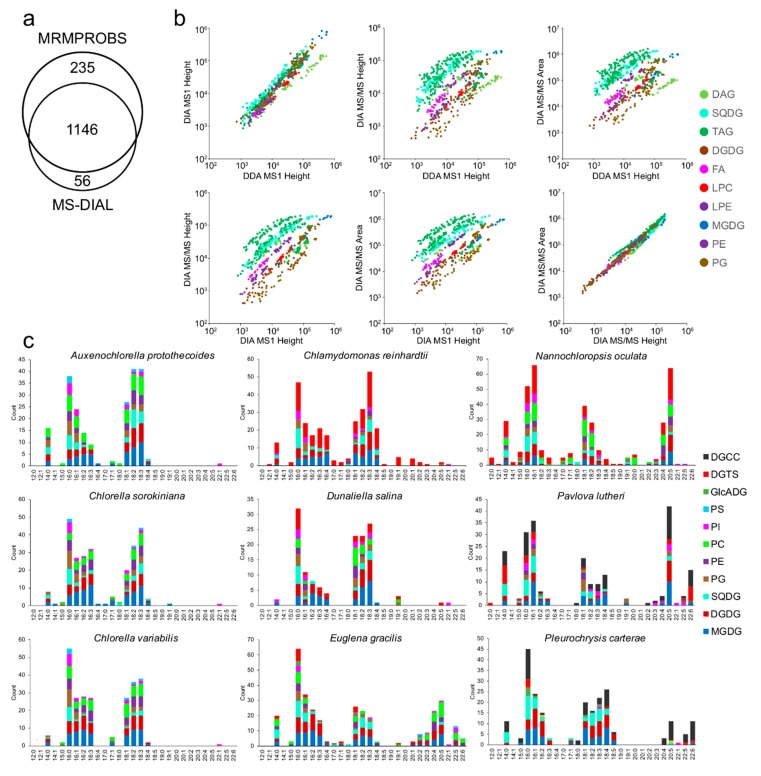
Summary of lipid annotations and comparison of MS^1^- and MS/MS-centric peak quantifications. (**a**) Counts of the annotated lipids in untargeted (MS-DIAL) and targeted (MRMPROBS) analyses. (**b**) The correlations of ion abundances where the MS^1^-centric peak heights in the DDA and DIA data and MS/MS-centric peak height and area are examined. (**c**) The statistics of acyl chains in all lipid classes except for triacylglycerol (TAG) for each algal species. The x- and y-axis shows the fatty acid information and frequency, respectively. The TAG lipids are not used for the counts because many combinations are observed in a single MS/MS spectrum, which may include false-positive identifications.

**Table 1 metabolites-09-00119-t001:** Summary of annotated lipids.

Super Class	Class	*Auxenochlorella protothecoides*	*Chlorella sorokiniana*	*Chlorella variabilis*	*Chlamydomonas reinhardtii*	*Dunaliella salina*	*Euglena gracilis*	*Nannochloropsis oculata*	*Pavlova lutheri*	*Pleurochrysis carterae*	Total
Fatty acids	FA	5	5	5	5	4	9	6	6	9	11
Glycerolipids	DAG	13	18	21	16	2	61	27	5	13	100
Glycerolipids	TAG	91	80	144	97	152	481	231	126	121	622
Phospholipids	PC	25	21	26	0	10	33	42	1	0	75
Phospholipids	PE	12	18	19	5	5	14	16	4	1	46
Phospholipids	PG	12	16	13	9	9	9	13	8	1	28
Phospholipids	PS	3	2	3	0	0	1	0	0	0	6
Phospholipids	PI	8	8	9	8	6	7	14	8	1	19
Phospholipids	PMeOH	0	0	0	0	0	29	0	0	0	29
Phospholipids	LPC	4	8	6	0	0	1	6	0	0	10
Phospholipids	LPE	2	6	2	1	0	0	3	0	0	8
Phospholipids	LPG	0	2	1	2	0	1	0	0	0	2
Phospholipids	LPS	0	1	1	0	0	0	0	0	0	1
Oxidized lipids	OxPC	0	0	3	0	0	0	0	0	0	3
Oxidized lipids	OxPE	0	0	3	0	0	0	0	0	0	3
Oxidized lipids	OxPG	0	2	6	0	1	0	0	0	0	7
Oxidized lipids	OxPI	0	0	3	0	0	0	0	0	0	3
Algal lipids	LDGTS/LDGTA	0	0	0	14	4	9	21	1	3	34
Algal lipids	DGTS/DGTA	0	0	0	68	12	12	68	14	4	134
Algal lipids	LDGCC	0	0	0	0	0	0	0	7	14	16
Algal lipids	DGCC	0	0	0	0	0	0	0	35	38	47
Plant lipids	MGDG	25	40	35	26	16	42	18	24	27	82
Plant lipids	DGDG	19	22	28	23	15	39	26	12	28	64
Plant lipids	SQDG	16	15	10	20	8	25	22	16	24	41
Plant lipids	GlcADG	0	0	0	0	0	3	0	1	4	5
Ceramides	Cer-AP	8	8	8	7	8	8	8	8	8	10
Ceramides	Cer-NP	8	7	8	7	6	6	10	7	6	10
Ceramides	Cer-NS	0	0	0	0	0	1	4	0	2	5
Ceramides	Cer-NDS	1	2	4	1	2	3	4	2	2	6
Ceramides	Cer-AS	1	1	1	1	1	3	1	2	1	4
Ceramides	Cer-ADS	0	0	3	0	0	1	0	0	0	4
Ceramides	HexCer-AP	0	0	0	0	0	0	0	2	0	2
Total		253	282	362	310	261	798	540	289	307	1437

The count of annotated lipids is provided for each lipid class and algal species. DGTS/DGTA indicates that these lipid classes are annotated using the same characteristic ions.

**Table 2 metabolites-09-00119-t002:** Diagnostic ions for lipid characterizations.

Lipid Class	Ion Mode	Adduct Type	Example	Diagnostic Ions (Lipid Class)	Diagnostic Ions (Acyl Chains)
LDGCC	Positive	[M+H]^+^	LDGCC 18:0	*m*/*z* 104.107 C_5_H_14_NO^+^, *m*/*z* 132.102 C_6_H_14_NO_2_^+^	-
DGCC	Positive	[M+H]^+^	DGCC 18:0-20:4	*m*/*z* 104.107 C_5_H_14_NO^+^, *m*/*z* 132.102 C_6_H_14_NO_2_^+^	NL of SN1 (*m*/*z* 538.374 C_30_H_52_NO_7_+), NL of SN1+H_2_O (*m*/*z* 520.363 C_30_H_50_NO_6_+), NL of SN2 (*m*/*z* 518.405 C_28_H_56_NO_7_+), NL of SN2+H_2_O (*m*/*z* 500.395 C_28_H_54_NO_6_+)
OxPC	Negative	[M+HCOO]^−^	OxPC 18:0-20:4+2O	NL of HCOO+CH3 (*m*/*z* 826.56 C_45_H_81_NO_10_P^−^)	SN1 (*m*/*z* 283.264 C_18_H_35_O_2_^−^), SN2 (*m*/*z* 335.223, C_20_H_3_1O_4_^−^), SN2−H_2_O (*m*/*z* 317.212 C_20_H_29_O_3_^−^) *SN2−2H_2_O (*m*/*z* 299.202 C_20_H_27_O_2_^−^)
OxPE	Negative	[M−H]^−^	OxPE 18:0-20:4+2O	*m*/*z* 196.038 C_5_H_11_NO_5_P^−^	SN1 (*m*/*z* 283.264 C_18_H_35_O_2_^−^), SN2 (*m*/*z* 335.223, C_20_H_31_O_4_^−^), SN2−H_2_O (*m*/*z* 317.212 C_20_H_29_O_3_^−^) *SN2−2H_2_O (*m*/*z* 299.202 C_20_H_27_O_2_^−^)
OxPG	Negative	[M−H]^−^	OxPG 18:0-20:4+2O	*m*/*z* 152.995 C_3_H_6_O_5_P^−^	SN1 (*m*/*z* 283.264 C_18_H_35_O_2_^−^), SN2 (*m*/*z* 335.223, C_20_H_31_O_4_^−^), SN2−H_2_O (*m*/*z* 317.212 C_20_H_29_O_3_^−^) *SN2−2H_2_O (*m*/*z* 299.202 C_20_H_27_O_2_^−^)
OxPI	Negative	[M−H]^−^	OxPI 18:0-20:4+2O	*m*/*z* 297.038 C_9_H_14_O_9_P^−^, *m*/*z* 241.012 C_6_H_10_O_8_P^−^	SN1 (*m*/*z* 283.264 C_18_H_35_O_2_^−^), SN2 (*m*/*z* 335.223, C_20_H_31_O_4_^−^), SN2−H_2_O (*m*/*z* 317.212 C_20_H_29_O_3_^−^) *SN2−2H_2_O (*m*/*z* 299.202 C_20_H_27_O_2_^−^)
PMeOH	Negative	[M−H]^−^	PMeOH 18:0-20:4	*m*/*z* 167.012 C_4_H_8_O_5_P^−^, *m*/*z* 110.985 CH_4_O_4_P^−^	SN1 (*m*/*z* 283.264 C_18_H_35_O_2_^−^), SN2 (*m*/*z* 303.233 C_20_H_31_O_2_^−^)
GlcADG	Negative	[M−H]^−^	GlcADG 18:0-20:4	*m*/*z* 249.062 C_9_H_13_O_7_	SN1 (*m*/*z* 283.264 C_18_H_35_O_2_^−^), SN2 (*m*/*z* 303.233 C_20_H_31_O_2_^−^)
LPG	Negative	[M−H]^−^	LPG 18:0	*m*/*z* 152.995 C3H6O5P^−^	SN1 || SN2 (*m*/*z* 283.264 C_18_H_35_O_2_^−^)
LPS	Negative	[M−H]^−^	LPS 18:0	NL of C3H6NO2 (*m*/*z* 437.267 C21H42O7P^−^)	SN1 || SN2 (*m*/*z* 283.264 C_18_H_35_O_2_^−^)

The product ion *m*/*z* or neutral loss information is summarized. In this study, LDGCC and DGCC are characterized in positive ion mode and the other lipid classes are characterized in negative ion mode. The example lipid molecule is used for the showcase of fragment ions. The term “20:4” describes the fatty acid moiety of lipids including 20 carbons and 4 double bonds. The asterisk denotes a minor fragment ion that is rarely detected under our experimental conditions.

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
