# Peer review of "Mass Spectrometry Data Repository Enhances Novel Metabolite Discoveries with Advances in Computational Metabolomics"

_metabolites, 2019, doi:10.3390/metabo9060119_

Round 1
Reviewer 1 Report
This manuscript nicely describes the value of public data repositories and the new information re-extraction they offer through data re-analysis using enriched databases and spectral libraries and the combination of informatic tools, like MS-DIAL and MRMPROBS in this case study. The strongest point of this manuscript is a demonstration of the strategy to feed the information obtained by untargeted data processing in MS-DIAL to MRMPROBS for subsequent targeted data analysis which result in higher number of annotated lipid species due to more specific MS/MS spectra resolution.
Although several specific examples are provided, I believe few following minor revisions would enhance the quality of this manuscript:
It remains unclear why the EICs of product ions cannot be integrated in MS-DIAL as they were already extracted and aligned with the respective precursor?
Was the lipid annotation following the automatized MS/MS matching against in silico spectral libraries manually inspected and curated? Authors should specify that the in silico spectral libraries do have important limitations - that the MS/MS matching can generate a high number of false positive hits and that the manual curation of data remains an obligation. Also, that they do perform well for specific classes of small molecules such as lipids (due to their specific fragmentation patterns per class - i.e. head groups, and fatty acid composition) however their succes is very limited with polar small molecules. It is important that authors do stress the added value of experimentally acquired spectral libraries (such as METLIN, mzCloud, growing MassBank, etc.). They can also specify that these libraries due to their added value (including the high cost in terms of time and expertise to acquire the data) are often not downloadable although open-access.
Authors should also clearly state that the annotation of additional lipid species has worked very well due to HILIC chromatography used for lipid class separation, otherwise the lipids from other classes (with same fatty acid composition) could coelute and significantly enhance the complexity of data deconvolution (and even make it impossible in some cases).
Author Response
We appreciated your comments and suggestions to improve our manuscript before publication. Please see the attached file to see our revision.

Reviewer 2 Report
The authors present an interesting case study where new findings are discovered by re-analysing previously published data. The case study is well described and the possibility to re-analyse old data thanks to public data repositories and new data opens a new field of interest for metabolomics, the computational metabolomics. However, the study seems to be an application of a specific software platform, while the topic suggested by the title is more large. I suggest to describe in more detail in a dedicated methodological section the main issues about data storage, in silico library building, data retrieval and so on, and the strategies used to solve them in order to introduce the reader in the field of computational metabolomics. Thus, the case study can be reported. A more general discussion about the importance to integrate targeted and untargeted data should be provided. Limits of the current technologies (hadware and sofware) should be discussed.
Author Response
We much appreciated your comments and suggestions. Please see the attached PDF file to read our point-by-point responses.

Round 2
Reviewer 2 Report
The authors have addressed all my points. In my opinion the manuscript can be accepted.